# TOTAL STYLE TRANSFER
# WITH A SINGLE FEED-FORWARD NETWORK

## ABSTRACT

Recent image style transferring methods achieved arbitrary stylization with input content and style images. To transfer the style of an arbitrary image to a content image, these methods used a feed-forward network with a lowest-scaled feature transformer or a cascade of the networks with a feature transformer of a corresponding scale. However, their approaches did not consider either multi-scaled style in their single-scale feature transformer or dependency between the transformed feature statistics across the cascade networks. This shortcoming resulted in generating partially and inexactly transferred style in the generated images. To overcome this limitation of partial style transfer, we propose a total style transferring method which transfers multi-scaled feature statistics through a single feed-forward process. First, our method transforms multi-scaled feature maps of a content image into those of a target style image by considering both inter-channel correlations in each single scaled feature map and inter-scale correlations between multi-scaled feature maps. Second, each transformed feature map is inserted into the decoder layer of the corresponding scale using skip-connection. Finally, the skip-connected multi-scaled feature maps are decoded into a stylized image through our trained decoder network.

## 1 INTRODUCTION

Recent image style transferring methodsJohnson et al. (2016); Ulyanov et al. (2016; 2017) improved image generating speed up to sub-realtime processing by learning a feed-forward network of a single style or several fixed stylesDumoulin et al. (2017). Huang et al.Huang & Belongie (2017) proposed an adaptive instance normalization layer (AdaIN) that adaptively transforms the statistics of an encoded content feature into that of a target style feature and they achieved style transferring into arbitrary input target style. However, they did not consider multi-scaled style characteristics of an imageGatys et al. (2016) but only a single scale feature in differentiating styles inside AdaIN layer. Li et al.Li et al. (2017b) proposed to use cascade networks that cumulatively transfer the multi-scaled style characteristics by using a network per scale as shown in fig.1 (a). They also transformed correlation between channels of feature map by using their whitening and coloring transformer (WCT). However, their cascade scheme requires multiple feed-forward passes to produce a stylized image and it is not guaranteed that the transferred style through a network is preserved after going through the subsequent networks because of inter-scale dependency in the multi-scaled styles of an image. Therefore, transferring multi-scaled style without interference between scales is still remained to study.

In this paper, we propose an improved feed-forward network structure (fig.1 (b)) and a multi-scaled style transferring method, called total style transfer, to efficiently perform style transfer in all scales of feature maps through a single feed-forward pass. Our work has the following contributions.

- Transforming both intra-scale and inter-scale statistics of multi-scaled feature map: There exist both of inter and intra-correlations in the encoded multi-scaled feature map as shown in fig.2 (b). Therefore, we match the second-order statistics, i.e., mean and covariance, of the encoded multi-scaled feature map considering the correlations not only between channels in each scale (intra-scale correlation) but also between scales (inter-scale correlation). Our feature transformer makes the transformed feature map closer to the target style feature map and this results in an output style closer to the target style.

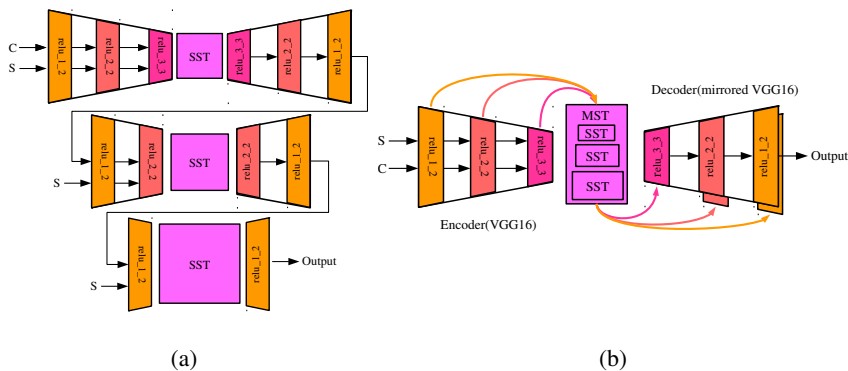

(a)            (b)

Figure 1: Network structure diagram for applying multi-scale stylization. (a) The cascade network scheme transfers style of an image into that of a target style image by using single scale transformers (SST) scale by scale. (b) Our multi-scaled style transformer (MST) transfers multi-scaled styles in a feed-forward process by using internal SSTs and a skip-connected decoder.

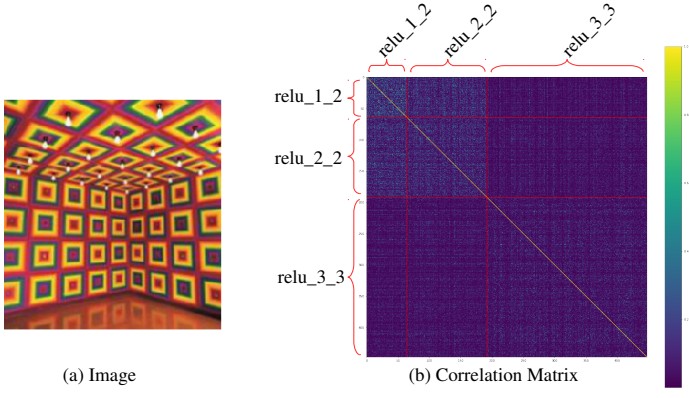

(a) Image            (b) Correlation Matrix

Figure 2: Correlation between channels in the multi-scaled feature map of the input image (a) extracted from the pre-trained VGG16 Simonyan & Zisserman (2014). The area corresponding to each scale of feature map is divided into red lines. In case of intra-scale feature transform, the diagonal rectangles on the correlation matrix are used. In case of inter-scale feature transform, entire region of the correlation matrix is considered.

- Decoder learning with multi-scaled style loss: we use a multi-scaled style loss consistent to the feature transformer, i.e., mean and covariance loss between the concatenated feature map (fig.3 (a)). Using our multi-scaled style loss allows the decoder network to generate an output image of co-occurring multi-scale patterns which is better style expression than independently occurring scale patterns on the image that the existing methods generated.

- Multi-scaled style transfer with a single feed-forward network: we use skip-connections for each decoder layer as shown in fig.1 (b) to consider the transformed feature map as well as the decoded feature map. By doing this, the style of scale corresponding to the layer and the transferred multi-scaled style so far are optimally merged into the next layer. Therefore, our method transfers multi-scaled style through a feed-forward pass in a single network instead of multiple feed-forward passes of cascade networks (fig.1 (a)) without considering inter-scale correlation.

In the remained of this paper, we review previous works closely related to this work in Sec. 2, our multi-scaled style transforming method is described in Sec. 3, the effectiveness of our method is tested and proven by a bundle of experiments in Sec. 4, and this work is concluded in Sec. 5.

## 2   RELATED WORK

Gatys et al. Gatys et al. (2016) represented content and style features of an image using a deep feature map, i.e., the filtered responses of a learned convolutional neural network (CNN). To stylize an input image, they performed pixel-wise optimization of the image to reduce the feature losses of

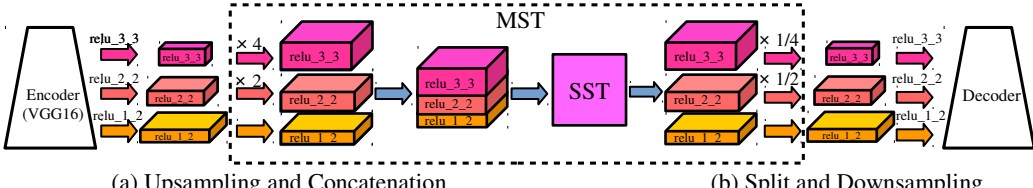

(a) Upsampling and Concatenation          (b) Split and Downsampling

Figure 3: Each diagram shows the process of merging multi-scale features and dividing them into the original size for inter-scale feature transform. (a) Merging multi-scale features is performed as upsampling each scale feature by using nearest neighborhood interpolation to the largest size followed by concatenating them along the channel axis. (b) After feature transform, the transformed feature is split into multi-scale features and downsampled into its original size. Each split feature is inserted into the decoder layer of the corresponding scale by using skip-connection

the input and a target style image. Their method can transform an image in any target style by a time consuming online optimization process. Li et al. Li et al. (2017a) interpreted the process of generating a stylized image by matching Gram matrix Gatys et al. (2016) as a problem of maximum mean discrepancy (MMD) specifically with a second-order polynomial kernel.

Using a feed-forward neural network Johnson et al. (2016); Ulyanov et al. (2016) moved the time consuming online optimization process into an offline feed-forward network learning to speed up the image generating speed of the previous method Gatys et al. (2016). The generated style quality was also improved by using instance normalization (IN) Ulyanov et al. (2017).

Dumoulin et al. Dumoulin et al. (2017) extended the previous single style network to transfer multiple styles. They used conditional instance normalization (CIN) layers in a single network. As selecting learnable affine parameters corresponding the specific style in the CIN layers, the feed-forward network transfers the selected style. This method achieved generation of pre-trained multiple styles with a single network.

To generalize a single network for arbitrary style transfer, Huang et al. Huang & Belongie (2017) proposed to use a feature transformer called adaptive instance normalization (AdaIN) layer between encoder and decoder networks. Once feature maps of content and target style images are encoded, AdaIN directly adjusts the mean and standard deviation of a content feature map into those of a target style feature map, and then the adjusted feature map is decoded into an output image of the target style. Li et al. Li et al. (2017b) further improved the arbitrary stylization by using covariance, instead of standard deviation, which considers the correlation between feature channels.

To transfer multi-scaled style, Li et al. Li et al. (2017b) used cascade networks, one scale of style transfer per network. Very recent feed-forward network approach Sheng et al. (2018) used multiple transformers in each scaled layer of their decoder network, where each transformer transfers corresponding scale of style and this reduced redundancy of cascade networks. However, their method did not consider the correlation between scales.

Recent generative adversarial networks (GANs) Isola et al. (2017); Zhu et al. (2017a;b) were used to transform a style per network as an application of their image translation. And there are also many methods dealt with various aspects of image style transfer such as photorealism Luan et al. (2017), spatial color control Gatys et al. (2017), and multi-style interpolation Huang & Belongie (2017); Dumoulin et al. (2017); Ghiasi et al. (2017). But, those methods did not deal with multiple scales of style.

## 3 METHOD

### 3.1 MULTI-SCALE FEATURE TRANSFORM

As described in Gatys et al. (2016), each scaled feature of CNN represents different style characteristics of an image. So, we utilize multiple feature transformers for each scale feature to transfer total style characteristics of an image. In this section, we explain two schemes of our total style transfer, i.e, intra-scale and inter-scale feature transform, with a single feed-forward network.

### 3.1.1 INTRA-SCALE FEATURE TRANSFORM

Our intra-scale feature transform is a set of independent single-scale style transforms as an extended multi-scale version of the single-scale correlation alignment of CORAL Sun et al. (2016) or WCT Li et al. (2017b).

Suppose we have i-th scale features $F_{c,i} \in R^{C_i \times (H_{c,i} \times W_{c,i})}$ of content image and $F_{s,i} \in R^{C_i \times (H_{s,i} \times W_{s,i})}$ of style image, where $C_i$, $H_{c,i}$ (or $H_{s,i}$), and $W_{c,i}$ (or $W_{s,i}$) represent the number of channels, spatial height, and width of i-th scale features respectively. For a single-scale style transform with these features, CORAL or WCT performs (1) style normalization and (2) stylization sequentially. In the style normalization step, first zero-centered feature $\bar{F}_{c,i} \in R^{C_i \times (H_{c,i} \times W_{c,i})}$ of the content feature $F_{c,i}$ is calculated and then the content feature $F_{c,i}$ is normalized into $\hat{F}_{c,i}$ by using its own covariance matrix $cov(\bar{F}_{c,i}) \in R^{C_i \times C_i}$ as eq.1.

$$\hat{F}_{c,i} = cov(\bar{F}_{c,i})^{-\frac{1}{2}} \cdot \bar{F}_{c,i}. \tag{1}$$

In the stylization step, the normalized content feature $\hat{F}_{c,i}$ is stylized into $F_{cs,i}$ by using the square root of covariance matrix $cov(\bar{F}_{s,i}) \in R^{C_i \times C_i}$ of zero-centered style feature $\bar{F}_{s,i}$ and spatial mean $\mu_{s,i} \in R^{C_i \times 1}$ of the style feature $F_{s,i}$ as eq.2.

$$F_{cs,i} = cov(\bar{F}_{s,i})^{\frac{1}{2}} \cdot \hat{F}_{c,i} + \mu_{s,i} \cdot \mathbf{1}^{1 \times (H_{s,i} \times W_{s,i})}. \tag{2}$$

Our intra-scale transform method applies the above single-scale transform independently to each scaled feature for $i = 1..3$ corresponding to {relu_1_2, relu_2_2, relu_3_3} layers. Then, those transformed features $F_{cs,i}$, $i = 1..3$ are inserted into the subsequent decoder through skip-connenction. More detail about skip-connection will be described in Sec.3.1.3.

### 3.1.2 INTER-SCALE FEATURE TRANSFORM

As shown in fig.2 (b), there exists not only inter-channel correlation in a certain scale feature but also inter-scale correlation between multi-scale features. These correlations should be considered in order to transfer total style characteristics of an image. CORAL Sun et al. (2016) or WCT Li et al. (2017b) did not consider inter-scale correlation but only inter-channel correlation. Therefore, we propose inter-scale feature transformer which considers more style characteristics of image for style transfer.

To perform feature transform considering both inter-channel and inter-scale correlations, we apply the intra-scale feature transform of Sec.3.1.1 to the concatenated feature $F'_c \in R^{\sum_i C_i \times (H_{c,1} \times W_{c,1})}$ of content image and $F'_s \in R^{\sum_i C_i \times (H_{s,1} \times W_{s,1})}$ of style image (eq.3) instead of independently applying to each scaled features $F_{c,i}$ and $F_{s,i}$ of Sec.3.1.1.

$$F'_c = \left[ F_{c,1}{}^T \ U(F_{c,2})^T \ U(F_{c,3})^T \right]^T, \ F'_s = \left[ F_{s,1}{}^T \ U(F_{s,2})^T \ U(F_{s,3})^T \right]^T. \tag{3}$$

As shown in fig.3 (a), the content features $F_{c,i}$ and style features $F_{s,i}$ for $i = 1..3$ are spatially upsampled into $U(F_{c,i})$ and $U(F_{s,i})$ of a common size (we use the largest size of $F_{c,1}$ or $F_{s,1}$ corresponding to {relu_1_2}) and concatenated into $F'_c$ and $F'_s$ respectively along the channel axis. After going through a transformer, the transformed feature $F'_{cs}$ is split and downsampled into $F'_{cs,i} \in R^{C_i \times (H_{c,i} \times W_{c,i})}$ of the original feature size as shown fig.3 (b) and eq.4,

$$F'_{cs,1} = F'_{cs}[1 : C_1], \ F'_{cs,i} = D_i(F'_{cs}[\sum_k^{i-1} C_k + 1 : C_i]) \ for \ i > 1, \tag{4}$$

where, $D_i(f)$ is a function which spatially downsamples $f$ into $H_{c,i} \times W_{c,i}$. These features are inserted into the subsequent decoder through skip-connenction of Sec.3.1.3.

### 3.1.3 SINGLE DECODER NETWORK WITH SKIP-CONNECTIONS

To utilize the transformed multi-scale features in generating output stylized image, decoding architecture of previous decoder network should be modified because each decoder layer has two input feature maps as fig.1 (b), one is a decoded feature map from the previous decoder layer, the other is a (intra-scale or inter-scale) transformed feature from the transformer. We adopt skip-connection, which has been applied to several applications of computer vision field Ronneberger et al. (2015); Mao et al. (2016); Isola et al. (2017); Bagautdinov et al. (2018) but not to image style transfer yet, to merge the two feature maps in decoding process as shown in fig.3 (b). Skip-connected two scale features are optimally merged by learnable convolution layer and this improves the quality of the decoded image by considering multi-scale filter responses.

Our method is different from the previous cascade scheme of Li et al. (2017b) because we use a single encoder/decoder network, parallel transformers for each scale feature, and merges multi-scaled

styles optimally while the cascade scheme needs several encoder/decoder networks (one network per scale feature) and sequentially transfers scaled styles from large to small scale at the risk of degradation in previously transferred scale of style. Avatar-Net Sheng et al. (2018) also used a single decoder like ours but it sequentially applied feature transformers from large to small scale without considering possible degradation of the previously transferred scale.

## 3.2 Multi-scaled style loss

We need an appropriate objective function for decoder network to generate a stylized image from the transformed feature map. Among the existing losses such as Gram Gatys et al. (2016), Mean-Std Huang & Belongie (2017), and reconstruction error Li et al. (2017b), we adopt Mean-Std loss Huang & Belongie (2017) with some modification because of its consistency with AdaIN transformer. Instead of using Mean-Std loss as it is, we use Mean-Covariance loss to additionally consider interchannel and inter-scale correlations, which is consistent with our feature transformers described in Sec.3.1.

In case of using intra-scale feature transform (Sec.3.1.1), our style loss (eq.5) is calculated as the summation of mean loss Huang & Belongie (2017) and covariance loss, i.e., square root of Frobenius distance between covariance matrices of feature maps of output and target style images. In case of using inter-scale feature transform (Sec.3.1.1), the summation of mean and covariance losses of the concatenated features are used as the style loss (eq.6).

$$L_{style_{intra}} = \sum_i ||\mu_{s,i} - \mu_{o,i}|| + \sum_i ||cov(\bar{F}_{s,i}) - cov(\bar{F}_{o,i})||, \qquad (5)$$

$$L_{style_{inter}} = ||\mu'_s - \mu'_o|| + ||cov(\bar{F}'_s) - cov(\bar{F}'_o)||, \qquad (6)$$

where subscript $o$ represents of output stylized image.

# 4 Experimental Result

## 4.1 Experimental Setup

We used VGG16 feature extractor Simonyan & Zisserman (2014) as the encoder and a mirror-structured network as the decoder of our style transfer network. Our decoder network has 2 times larger number of channels in the corresponding layer of skip-connections than the previous methods Dumoulin et al. (2017); Huang & Belongie (2017). {relu_1_2, relu_2_2, relu_3_3, relu_4_3} layers were used in calculating style loss and {relu_3_3} layer in calculating content loss. Here, we used the same content loss of Gatys et al. (2016) and our multi-scaled style loss in Sec.3.2. For training data set, MS-COCO train2014 Lin et al. (2014) and Painter By Numbers Kingma & Ba (2015) were used as content image set and large style image set respectively. Each dataset consists of about 80,000 images. And we used an additional small style image set of 77 style images to verify the effect of our proposed method as the number of training style increases. Each image was resized into 256 pixels in short side maintaining the original aspect ratio in both training and test phases, and, only for training phase, randomly cropped into (240, 240) pixels to avoid boundary artifact. We trained networks with 4 batches of random (content, style) image pairs, 4 epochs, and learning rate of $10^{-4}$. And all experiments were performed on Pytorch 0.3.1 framework with NVIDIA GTX 1080 Ti GPU card, CUDA 9.0, and CuDNN 7.0.

## 4.2 Comparison of inter-scale and intra-scale feature transforms

In order to verify the effect of our multi-scale feature transform for varying number of training style images, we trained two networks, one with the small style image set of 77 images and the other with the large style image set of about 80,000 images. Then we compared the output stylized images of the networks.

Fig.4 shows an example of the output stylized images using our intra-scale or inter-scale feature transform method. With the network trained by a small style image set, the result images generated by our intra-scale transform (fig.4 (c)) show very similar texture style to the target style images (fig.4 (b)). And those by our inter-scale transform (fig.4 (d)) show even better style of texture. With the network trained by a large style image set (fig.4 (e,d)), the result images also show the same tendency that inter-scale is better in expressing the texture of target style. Because of the existing correlations between scales as shown in fig.2 (b), inter-scale feature transform which considers inter-scale correlations shows the better quality of style than intra-scale transform.

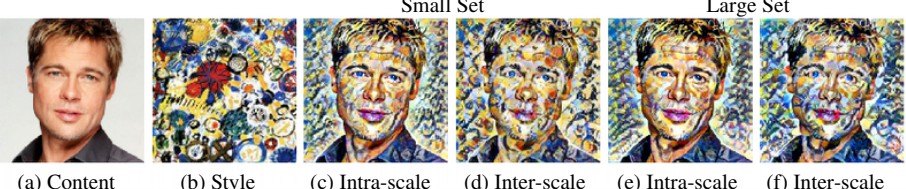

(a) Content (b) Style (c) Intra-scale (d) Inter-scale (e) Intra-scale (f) Inter-scale

Figure 4: Comparison of intra-scale and inter-scale transform: (a) input content image, (b) target style, (c,d) result of each model learned small style set and (e,f) results of learning large style set.

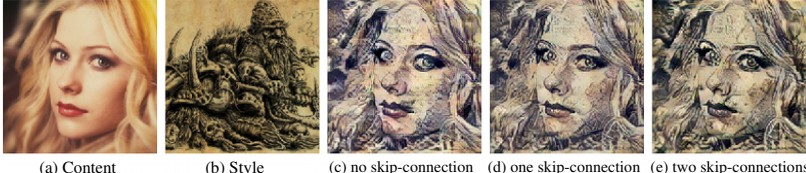

(a) Content (b) Style (c) no skip-connection (d) one skip-connection (e) two skip-connections

Figure 5: Output stylized images according to the number of skip-connections: The (content, style) losses of output images are (c) (0.910, 0.535), (d) (0.707, 0.512), and (e) (0.814, 0.497). The more skip-connections are used, the less style loss is acheived. And there is a trade-off between style and content losses.

### 4.3 ANALYSIS OF SKIP-CONNECTED DECODER NETWORK

To verity the effect of skip-connections in our style transfer network, we trained three different networks. The first one has the conventional single layer encoder/transformer/decoder architecture Huang & Belongie (2017); Li et al. (2017b) which a sinlge feature transformer on $\{relu\_3\_3\}$ without skip-connection. The second one has muti-scale feature transformers on $\{relu\_3\_3, relu\_2\_2\}$ and one skip-connection on $\{relu\_2\_2\}$. The last one has multi-scale feature transformers on $\{relu\_3\_3, relu\_2\_2, relu\_1\_2\}$ and two skip-connections on $\{relu\_2\_2, relu\_1\_2\}$.

Fig.5 shows an example of the output stylized image from the three different networks. As the number of skip-connections increases from fig.5(c) to (e), the style loss decreases from 0.535 to 0.497 and, accordingly, the color tone of the stylized image is getting better matched to the target style (fig.5(b)) and small patterns appear.

To clarify the contributions of the skip-connected feature from the transformer and the decoded feature from the previous scale of decoder layer, we observed the absolute values of loss gradients with respect to the convolution weights of skip-connected decoder layers during the network learning process. As shown in fig.6(a), the gradient values for the skip-connected feature (channel indices from 129 to 256) on $\{relu\_2\_2\}$ layer of decoder network are much larger than those for the decoded feature (channel indices from 1 to 128) at the beginning of training. This means that the skip-connected feature which already has target style through transformer dominantly affected to the decoder learning at the start of training phase. This happens because the previous decoder layer $\{relu\_3\_3\}$ has random initial weights and outputs noisy feature at the start of training phase. As iteration goes, the gradient values for both features became similar to each other. This means that both skip-connected feature and decoded feature were equally utilized to generate an image of multi-scaled style. Fig.1(b) shows that the gradient values for the skip-connected feature (channel indices from 65 to 128) are smaller than those for the decoded feature (channel indices from 1 to 64) at the latter decoder layer. This means that the decoded feature of the latter decoder layer already has accumulated multi-scaled styles by the previous skip-connection and this resulted in the less impact of the skip-connected feature. However, using the skip-connection with the stylized feature of smaller scale has a certain effect on the result image in color tone matching as shown in fig.5(d,e).

### 4.4 COMPARING WITH THE EXISTING METHODS

#### 4.4.1 COMPARISON IN QUALITY OF OUTPUT STYLIZED IMAGE

We compared the image quality of our method with those of the existing methods Gatys et al. (2016); Huang & Belongie (2017); Li et al. (2017b). We took the output stylized images for Gatys et al. (2016) after 700 iterations of Adam optimization with learning rate $10^{-1}$, for Huang & Belongie (2017) with the same setting of our method except transformer and loss, and for Li et al. (2017b) with style strength $\alpha = 0.6$ (as mentioned in their work) and 3 cascade networks of VGG16 structure.

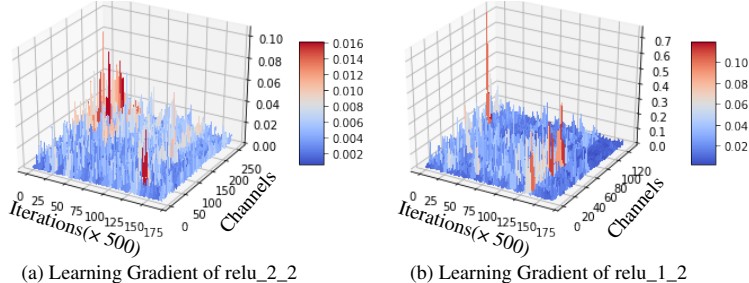

(a) Learning Gradient of relu_2_2     (b) Learning Gradient of relu_1_2

Figure 6: Amplitude of loss gradients with resprct to the convolution weights in the skip-connected decoder layers during the learning process: The gradients are drawn every 500 iterations. The former half of the channels are for decoded feature from the previous scale and the latter are for skip-connected feature from transformer. Based on the gradients of 1st skip-connected layer (a) and 2nd skip-connected layer (b), the skip-connected (transformed) feature highly seems to affect to the decoder in initial interations but both decoded and transformed features samely affect as iteration goes. And the latter decoder layer (b) is less affected by the skip-connected feature than the former layer (a).

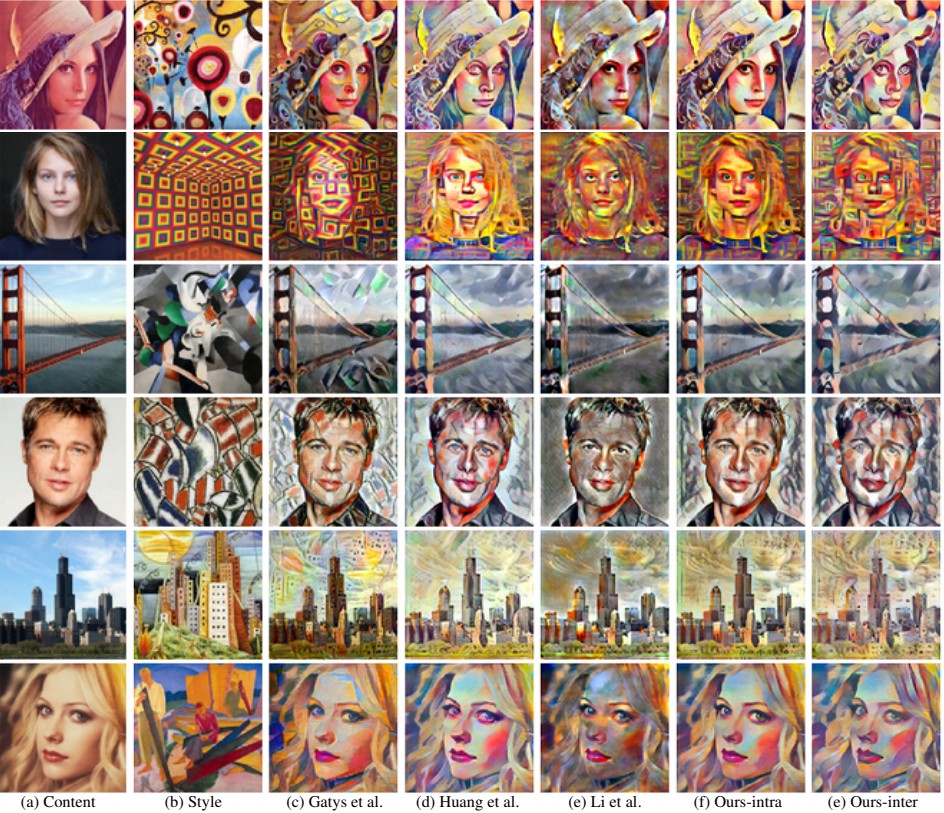

(a) Content  (b) Style  (c) Gatys et al.  (d) Huang et al.  (e) Li et al.  (f) Ours-intra  (e) Ours-inter

Figure 7: Output stylized images of the existing methods and our method: The target style images are not used for network learning.

Fig.7 shows the generated images from the existing and our intra/inter-scale feature transform methods. Compared to the online optimization method Gatys et al. (2016) (fig.7(c)), the other methods based on feed-forward network generated images of somewhat degraded style quality (fig.7(d e)). However, thanks to our muti-scaled style transfer which considering inter-channel or inter-scale correlation, texture detail and color tone of the generated image of our method with inter-scale (fig.7(f)) or intra-scale feature transform (fig.7(g)) are more similar to the target style than those of single-scale style transfer without considering inter-channel correlation Huang & Belongie (2017) (fig.7(d)) are.

| | On-line Learning | Trained with Small Style Image Set | | | | Trained with Large Style Image Set | | | |
|---|---|---|---|---|---|---|---|---|---|
| | (a) | (b) | (c) | intra | inter | (c) | (d) | intra | inter |
| Content Loss | 9.94 (6.23) | 14.80 (6.90) | 9.17 (3.06) | **8.72** **(4.41)** | 13.11 (5.02) | 8.69 (2.50) | 9.82 (2.07) | **8.62** **(3.81)** | 12.13 (4.23) |
| Inter-scale Covariance and Mean | 0.30 (0.10) | **0.42** **(0.19)** | 0.65 (0.31) | **0.55** **(0.24)** | **0.45** **(0.22)** | 0.73 (0.41) | 0.84 (0.42) | **0.622** **(0.33)** | **0.54** **(0.31)** |
| Inter-scale Gram | 0.11 (0.07) | **0.26** **(0.31)** | 0.78 (0.98) | **0.55** **(0.66)** | **0.39** **(0.53)** | 0.97 (1.52) | 1.13 (1.60) | **0.72** **(1.10)** | **0.58** **(0.94)** |
| Intra-scale Covariance and Mean | 0.32 (0.13) | **0.42** **(0.22)** | 0.72 (0.36) | **0.56** **(0.26)** | **0.49** **(0.25)** | 0.81 (0.47) | 0.86 (0.46) | **0.63** **(0.35)** | **0.57** **(0.34)** |
| Intra-scale Gram | 0.14 (0.13) | **0.34** **(0.51)** | 1.32 (1.74) | **0.81** **(1.12)** | **0.64** **(0.95)** | 1.70 (2.97) | 1.80 (2.88) | **1.06** **(1.85)** | **0.88** **(1.61)** |
| Mean, Std | 0.13 (0.08) | **0.24** **(0.18)** | 0.65 (0.37) | **0.45** **(0.27)** | **0.34** **(0.26)** | 0.75 (0.53) | 0.91 (0.57) | **0.52** **(0.35)** | **0.42** **(0.33)** |

Table 1: Average (standard deviation) losses of various image style transferring methods. Each methods (a) is Gatys et al. (2016), (b) is Dumoulin et al. (2017), (c) is Huang & Belongie (2017) and (d) is Li et al. (2017b).

Compared to Li et al. (2017b), the generated images of our method present the styles closer to the target styles because our methods trained a decoder network to minimize style loss between target style and output images while Li et al. (2017b) trained its decoder network to minimize reconstruction loss of content images.

We also compared our method to Avatar-Net Sheng et al. (2018) that performs multi-scale feature transform with a single decoder network. For a fair comparison, as the structure of Avatar-Net Sheng et al. (2018), we used VGG19 network Simonyan & Zisserman (2014) up to {relu_4_1} layer as the encoder and its mirrored structure as the decoder, and we also used additional style loss (eq.5 or eq.6) on image-level which is corresponding to the image reconstruction loss of Avatar-Net. As shown in fig.8, Both our intra (e) and inter (f) methods generated the stylized images with both detailed shapes of content images (a) and multi-scaled strokes of target style images (b). In contrast, the generated images of Avatar-Net (c, d) show somewhat deemed content shapes and blurred or burnt color patterns without detailed strokes. And selecting appropriate patch size corresponding to the scale of style pattern was necessary in Avatar-Net (in second row of fig.8, {patch size=1} (c) did not show large square patterns but {patch size=3} (d) did.) while our method did not require any scale matching parameter due to multi-scaled skip-connections in the decoder network.

### 4.4.2 COMPARISON IN CONTENT AND STYLE LOSSES

For a quantitative comparison, we compared the content and style losses of the generated images of the existing methods and ours.

Table.1 shows the measured average (standard deviation) losses across 847 tests of style transfer. Online optimization method Gatys et al. (2016) achieved the smallest style loss with a low content loss. Among the feed-forward networks trained with a small style image set, Dumoulin et al. (2017) achieved the lowest style loss (red colored numbers). This is because it has a learnable transformer and this resulted in a most optimized transformer. However, it is not extendable to arbitrary style transfer. Among the arbitrary style transferring methods, our method achieved the lowest style loss (blue colored numbers) with inter-scale feature transformer, and the second lowest style loss (green colored numbers) with intra-scale feature transformer. Among the feed-forward networks trained with a large style image set, our method shows the lowest style loss with inter-scale feature transformer and the second lowest style loss with intra-scale feature transformer in the same manner of the result with a small style image set.

For the content loss with large style image set, the best method in style loss (our-inter) shows the highest content loss and the second best method (our-intra) shows the lowest content loss. This interesting result can be interpreted that inter-scale correlation has not only style of an image but also content of the image. Tasks of transferring style and preserving content are a trade-off in feed-forward network methods. The result of Dumoulin et al. (2017) with small style image set also

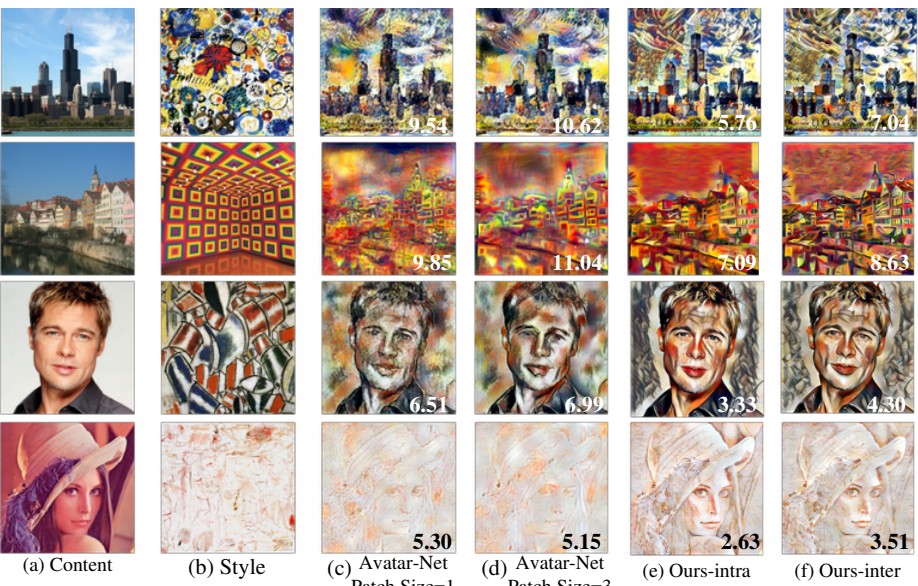

(a) Content    (b) Style    (c) Avatar-Net Patch Size=1    (d) Avatar-Net Patch Size=3    (e) Ours-intra    (f) Ours-inter

Figure 8: Comparison of single-network-multi-scale transform methods (Avatar-Net Sheng et al. (2018) and Ours): the numbers in bottom-right of images are content loss values.

shows the same content/style trade-off. Therefore, we can select either inter-scale or intra-scale feature transformer according to user preference or purpose of application.

### 4.4.3 COMPARISON IN PROCESSING SPEED AND MEMOCY EFFICIENTY

Our method achieved 31% less encoder/decoder feed-forward time (4.4 ms in average of 1000 trials with images of 240 by 240 pixels) and 4% less number of parameters (3,655,296 parameters) than the existing cascade network scheme Li et al. (2017b) (6.4 ms, 3,769,856 parameters).

## 5 CONCLUSION

In this paper, we proposed a total style transfer network that generates an image through a single feed-forward network by utilizing multi-scale features of content and style images.

Our intra-scale feature transformer transfers multi-scale style characteristics of the target style image and our inter-scale feature transformer transfers even more style characteristics of inter-scale correlation into the content image. By using our intra/inter scale feature transform, our total style transfer network achieved the lowest style loss among the existing feed-forward network methods.

In addition, we modified the feed-forward network structure by using skip-connections which make our decoder network to utilize all transformed multi-scale features. This modification allowed a single feed-forward network to generate image of multi-scaled style without using multiple feed-forward networks of cascade scheme, and resulted in the reduced test time by 31% and memory consumption by 4% compared to cascade network scheme.

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
