# OpenReview forum: "Total Style Transfer with a Single Feed-Forward Network"
_ICLR.cc/2019/Conference_

### Official Review · AnonReviewer1 · 2018-11-01
**Interesting architectures with limited novelty, also lack of clear presentation**

**Rating:** 4
**Confidence:** 3

**Review:**

This work proposes to use a single feed-forward network with two types of multi-scale transformers (MST) for image style transfer. The first transformer cascades existing single-scale transforms (SSTs), and the second one applies SST to the stacked feature maps. Skip connection is used between the MSTs and the decoder.

Pros

- Has quantitative evaluation
Table 1 includes quantitative results for different approaches, which is essential for proper evaluation.

Cons

- The problem is not very well motivated and the novelty is limited.
Why shall we care about multi-scale style transformation? Style transformation is about modifying an image to match certain style WITHOUT completely destroy its content. Allowing low level details to interfere high level content seems to be a bad idea.
As for novelty, it is claimed that this work is the first to use skp connection. However, Avatar-Net uses skip connection before.

- Empirical results not significant
In Table 1, why is (d) missing for the small set and (b) missing for the large set? It seems that (a) and (b) are already very good. Then why do we need the proposed methods? It is said that (b) is not extendable to arbitrary style transfer. How so? How do you define "arbitrary style transferring methods"?
I do not see how Fig.6 can tell us useful information about the skip connections. The corresponding paragraph in Sec.4.3 is not very convincing or informative.

- Writing can be improved.
In terms of content, terminologies are used without clear definitions. For example, what is "inter-scale dependency" in the introduction and what does "merges multi-scaled styles optimally" mean in Sec.3.1.3 (what is the optimality here)? It is confusing what corresponds to "intra-scale" or "inter-scale" throughout the paper. For example, the direct connection between relu_3_3 to relu_2_2 of the decoder in Fig.1b can also be interpreted as "inter-scale". As another example, "4 batches of random image pairs" in the experiment, do you mean a batch of 4 random pairs?
In terms of presentation, the grammar needs more careful checking. For examples, "in the remained of this paper", "each methods", etc. The meaning of the transpose in Eq.(3) is not clear. How do we transpose 3-dimensional tensor F? Also, know the difference between \citet and \citep and when to use them.

Minors
- In Eq.(4), according to the definition of C_i, for i > 1, the index should be (\sum_k C_k + 1):(\sum_k C_k + C_i)
- Above Eq.(5), "inter-scale feature transform (Sec.3.1.1)" should be Sec.3.1.2.
- In Fig. 7, the last column should be (g) instead of (e).

---

### Official Review · AnonReviewer2 · 2018-11-02
**Good ideas, but insufficient experimental validation**

**Rating:** 5
**Confidence:** 5

**Review:**

SUMMARY
The paper is concerned with the problem of arbitrary feed-forward style transfer, where a feed-forward model receives a content image and a style image as input, and must produce as output an image matching the content of the former and the style of the latter. The approach roughly follows that of [Li et al, NIPS 2017]: An encoder network (VGG pretrained on ImageNet) is used to extract features from both the style and content image; the features from the content image are adjusted to match the statistics from the style image, and the adjusted features are passed to a decoder network which generates the output image.

Compared to prior work, the main innovations are:
- Considering correlations between features from different encoder layers rather than only correlations within a single layer in both the feature adjustment step as well as in the loss
- An improved encoder / decoder architecture which uses skip connections in the decoder, allowing for a single encoder / decoder pair rather than a cascade of encoder / decoder pairs for each layer.

PROS
- Considering correlations between features from different encoder layers is a good idea
- The improved encoder / decoder architecture is significantly more efficient than the cascaded approach of [Li et al, NIPS 2017]

CONS
- Somewhat incremental
- Limited experimental evaluation
- Qualitative results not clearly better than existing methods
- Missing citation for multi-scale losses

LIMITED EXPERIMENTAL EVALUATION
One of the key claims of the paper is that “our method with inter-scale
(fig.7(f)) or intra-scale feature transform (fig.7(g)) are more similar to the target style than those of single-scale style transfer without considering inter-channel correlation” (Figure 7 caption); this claim is substantiated primarily by qualitative results in Figures 4, 7, and 8. Personally I don’t find the results with inter-feature correlations to be much better than those with only intra-feature correlations or the results from prior work. All recent style transfer methods depend on a host of hyperparameters like style and content weight, learning schedule, etc; in my experience differences in these hyperparameter settings can have large effects on the qualitative appearance of the generated images, and by varying these hyperparameters it is common to see qualitative differences similar to those shown in Figure 4 and 5. From the small number of qualitative results presented I do not think that the benefits of inter-scale correlations have been clearly demonstrated.

I appreciate that the authors tried to quantify their results by comparing loss values (Table 1) but unfortunately it’s hard to know how much the values of different losses correlate with human judgement of style quality.

In addition to selected qualitative results I would have liked to see a user study demonstrating human preference for images generated using inter-scale correlation losses, ideally across a range of different hyperparameter settings for each method.

GATYS BASELINE
From Table 1, the proposed method with intra-scale features achieves lower content loss than the direct optimization baseline (a) from [Gatys et al.] This is very surprising to me - typically direct optimization leads to much lower losses than any feedforward methods. From Section 4.4.1 this baseline uses Adam; in my experience using L-BFGS tends to achieve lower losses which may explain the results from Table 1.

MISSING CITATION FOR MULTI-SCALE STYLE TRANSFER
Instead of computing correlations between features from different encoder layers, [Wang et al, CVPR 2017] define a loss that considers the generated and style image at multiple spatial scales. This should be discussed in relation to the proposed method.

ENCODER / DECODER DETAILS
There are some missing details about exactly how the encoder and decoder are initialized and trained. I assume that the encoder was pretrained on ImageNet; is it updated during training or kept fixed? Is the decoder initialized randomly or does it mirror the pretrained ImageNet weights?

TYPOS / FORMATTING
There are minor typos throughout, e.g. “verity”, “sinlge” in Section 4.3, Paragrah 1. I also found the citation style to be somewhat jarring, especially in the introduction; parenthetical citations and better spacing may improve readability.

OVERALL
On the whole I feel that the paper is somewhat incremental. The inter-scale loss seems intuitively like a good idea, but I don’t think the paper presents sufficient experimental evidence to justify it. On the other hand the proposed decoder architecture seems like a clear improvement over the cascaded approach from [Li et al, NIPS 2017], as it is significantly more efficient without sacrificing quality. However I don’t feel that this alone is enough novelty for ICLR, so I lean slightly toward rejection.

---

### Official Review · AnonReviewer3 · 2018-11-03
**Interesting extension but lack of meaningful evaluation**

**Rating:** 4
**Confidence:** 5

**Review:**

The submission proposes a new model for the task of fast style transfer with arbitrary input images.
The model combines the existing approach of Li et al. 2017 [1] and the idea of inter-layer correlations into a new model architecture for fast style transfer. Notably the idea of inter-layer correlations is not new, it was first proposed by Novak and Nikulin [2] and recently picked up by Yeh and Tang[3] (both methods are not referenced in the present submission).
To evaluate the model, visual comparisons with other methods are shown and quantitative loss values for content and style loss are computed.

My main issue with the submission is that the scope of the evaluation is not suited to demonstrate a clear advantage of the proposed method over existing work. The qualitative comparison remains highly subjective. Based on figure 7 I could not tell which method performs the best style transfer and this might well be because the systematic perceptual differences between the methods are small.
To demonstrate a clear advance of the method in style transfer, I do believe it is necessary to run a user study showing that users systematically prefer the output of the proposed method over that of competing methods.

Additionally the quantitative evaluation leaves several open questions:
- Why is there no comparison with Ghiasi et al. 2017 [4]? This is the natural extension of Dumoulin et al. 2017 [5] for arbitrary fast style transfer.
- What exactly is the content and style loss that these methods are compared on? Is it the one used for the models in this submission? If so, were the compared methods also trained to minimise this loss function? If not this does not seem to be a fair comparison.
- Even if all methods were trained to minimise the same loss function, without user study it is unclear if that loss function is a good approximation of perceptual preference, which is the actual underlying target that style transfer aims to optimise.

As a side note, the provided baseline of the Gatys et al. [6] method is somewhat misleading because it is using the Adam optimiser. The original work used LBFGS to optimise the loss function and it is fairly well known that the choice of optimiser can have significant impact on the quality of the results.
I do not think this comparison is critical for the submission because the proposed method mainly competes with other fast style transfer methods, but I would recommend more care to be taken when reproducing existing work.

All in all I think the submission proposes an interesting combination of existing methods leading to reasonable extension of the body of work in fast style transfer. However, given the lack of meaningful evaluation I am not convinced that the proposed method substantially advances the field of fast style transfer to warrant publication at ICLR.


[1] Yijun Li, Chen Fang, Jimei Yang, Zhaowen Wang, Xin Lu, and Ming-Hsuan Yang. Universal style transfer via feature transforms. In Advances in Neural Information Processing Systems, pp. 385– 395, 2017b.
[2] Improving the Neural Algorithm of Artistic Style. R Novak, Y Nikulin - arXiv preprint arXiv:1605.04603, 2016 - arxiv.org
[3] Improved Style Transfer by Respecting Inter-layer Correlations MC Yeh, S Tang - arXiv preprint arXiv:1801.01933, 2018 - arxiv.org
[4] Golnaz Ghiasi, Honglak Lee, Manjunath Kudlur, Vincent Dumoulin, and Jonathon Shlens. Explor- ing the structure of a real-time, arbitrary neural artistic stylization network. In British Machine Vision Conference (BMVC), Sep 2017.
[5] Vincent Dumoulin, Jonathon Shlens, and Manjunath Kudlur. A learned representation for artistic style. In International Conference on Learning Representations (ICLR), Apr 2017.
[6] Leon A. Gatys, Alexander S. Ecker, and Matthias Bethge. Image style transfer using convolutional neural networks. In The IEEE Conference on Computer Vision and Pattern Recognition (CVPR), June 2016.

---

### Public Comment · ~Daan_Wynen1 · 2018-10-23
**Natural progression of WCT, but with issues and questions**

The overall idea of this paper is to improve upon WCT (a.k.a. "universal style transfer", Li et al. 2017b) by removing the need for successive encoding/decoding steps, thus performing style transfer in one single forward pass through an encoder/decoder pair, with the feature whitening and coloring happening in-between. As a natural second contribution, the whitening/coloring step is extended to take into account inter-layer covariance. The decoders are trained taking into account a style loss term.

My questions and comments are these:

1.
I want to start with the most obvious comment to make about this paper: the writing needs serious work. That is in terms of orthography, grammar, and wording. Nothing that cannot be done with the help of a (near) native, but to make this publishable, I think it needs to be done, because as it stands, it makes reading the paper much harder than it needs to be.
On top of that, the resolution of the figures you present is much too low. Using pdfimages and measuring the size of the results shows that the stylization results are presented at 90x90 px, while you write that stylization is performed at 256x256 px. This doesn't really allow to judge the quality of the results.

2.
The idea of using inter-layer covariance for style transfer is not actually new. The arxiv submission " Improved Style Transfer by Respecting Inter-layer Correlations", although to my knowledge not published elsewhere, does apply WCT to inter-layer gram matrices already. Since I don't know the identity of the authors, this might actually turn out to be a non-issue. Otherwise though, this paper would not be complete without mentioning said preprint.

3.
In section 4.4.1 you write that you use "3 cascade networks of VGG16 structure" for (Li et al. 2017b).
The original WCT used a different setup though, namely VGG-19, with the layers relu_X_1, X=1...5.
Does that mean that you also trained the decoders used for these experiments to invert the selected VGG-16 layers? Are those the same layers you use for your own method, i.e. relu_1_2, relu_2_2, relu_3_3? If so, could you explain your reasoning? Since you're building on Li et al. 2017b, why did you not use the same architecture and set of layers?

4.
The results using Li et al. in Figure 7 seem worse than expected.
Using the pytorch implementation from [1] I tried reproducing the last row, using the same content and style images found online [2, 3].
I used --fineSize 256 (rescaling to 256x256 pixels) and --alpha 0.6
With a truncated list of encoders/decoders to relu_[123]_1 I get a much better result than the one shown in Figure 7.
This would imply that the bad quality of the WCT results stems not from the method, but from your choice of target layers or from decoders that don't preserve the necessary information.
In fairness, with the full five encoding/decoding steps the result is barely recognizable.
So it would be good to elaborate on how these results i column (e) were generated.

5.
In section 4.3 you analyze the gradient values in the decoding layers that have skip connections as inputs.
This is with the goal of evaluating if de decoders make use of the skip connections, or if they rely mostly on the lower resolution outputs of the preceding layers.
You claim that since the gradient values equalize between the two groups of input channels, "both skip-connected feature and decoded feature were equally utilized", yet that would only be true if the resulting weights (and not just their gradients) were at least similar in magnitude, no?
Regarding figure 6: it is not clear from the caption what vertical axis and color are representing. This figure might be made much more readable, e.g. by aggregating the two groups of channels.

Overall, I like the idea, and would really like this to work and be published, since I would want to use it myself. The approach drops the property of WCT being "learning free" in that the decoders don't get trained on artworks at all, but that in itself is not a feature, just a charming peculiarity. Training decoders specifically for the task of generating credible artworks makes sense in my opinion. However, the issues mentioned above would have to be addressed.

[1] https://github.com/sunshineatnoon/PytorchWCT
[2] http://wallpaperswide.com/download/avril_lavigne_74-wallpaper-1680x1050.jpg
[3] https://commons.wikimedia.org/wiki/File:Pravka_pil-1927.jpg

---

> ### Author Response · Authors · 2018-10-29
> **Answers for the questions**
>
> Thank you for your paying attention to our manuscript and giving interesting questions.
>
> 1. We agree with you that our manuscript still needs to be improved in grammar and expression. As you mentioned, the figures also need to be improved in resolution. We will improve those errors in our final manuscript.
>
> 2. Thank you for introducing a method [1] that is also using inter-layer correlation. After inspecting the method [1], we can say that our method has critical differences from [1]. First, [1] uses only two neighboring layers but our method uses total layers to perform WCT. This means that our WCT transforms the whole correlation between all layers shown as figure 2(b) while [1] transforms partial diagonal regions of figure 2(b). Second, because of the first difference, [1] still uses decoder cascade to apply their partial correlation transformer to all layers at the risk of degrading the transformed feature of the previous decoder. But our method uses only a single decoder network and prevent degradation of the transformed features through skip-connections.
>
> 3. We adopted VGG16 and its mirrored structure in our encoder/decoder networks. This network configuration is based on the usage of VGG16 of the first feedforward image style transfer [2] and also for efficiency in experiments (VGG16 has much fewer parameters to optimize than VGG19). And for a fair comparison without the effect of different network configurations, we also used VGG16 network configuration for Li et al. 2017b [3] and other methods as described in the section of Experimental Setup.
>
> 4. As [3] uses VGG19 (with original layer configuration or even with our layer configuration relu_[1,2,3]_1), the result images of [3] have the different quality from our result images of figure 7(e) generated by using VGG16. We are also aware of the higher style quality of the generated images with VGG19 configuration. However, we did not add the experimental results of VGG19 configuration because of the page limitation of the manuscript. Instead, we added the result of Avatar-Net [5] with VGG19 configuration (figure ) where Avatar-Net is the most recent method and known as the better style quality than Universal (Li et al. 2017b [3]). We have a plan to add results of VGG19 configuration as a supplementary material when submitting a final version.
>
> 5. The convolution weights can have negative or positive values. Accordingly, even when the absolute values of weights are large, the output of the convolution layer can also be very small. Therefore, the absolute values of convolution weights are not appropriate to measure the effects of the decoded feature and skip-connected feature in generating a stylized image. Instead, we decided to measure the effectiveness of the two kinds of features (decoded feature and skip-connected feature) by observing the gradient (dL/dw) of loss (L) with respect to convolution weight (w) in training phase. In figure 6, the vertical axis and color represent the same quantity, i.e., the amplitude of the gradient values. For better readability, we will consider presenting the group amplitudes for the two groups of channels.
>
> [1] Yeh, Mao-Chuang, and Shuai Tang. "Improved Style Transfer by Respecting Inter-layer Correlations." arXiv preprint arXiv:1801.01933 (2018).
> [2] Johnson, Justin, Alexandre Alahi, and Li Fei-Fei. "Perceptual losses for real-time style transfer and super-resolution." European Conference on Computer Vision. Springer, Cham, 2016.
> [3] Li, Yijun, et al. "Universal style transfer via feature transforms." Advances in Neural Information Processing Systems. 2017.
> [4] https://github.com/sunshineatnoon/PytorchWCT
> [5] Sheng, Lu, et al. "Avatar-Net: Multi-scale Zero-shot Style Transfer by Feature Decoration." Proceedings of the IEEE Conference on Computer Vision and Pattern Recognition. 2018.

---

### Meta-Review · Area_Chair1 · 2018-12-04
**decision**

**Confidence:** 5
**Recommendation:** Reject

**Metareview:**

The reviewers and this AC agree that the paper is not of acceptable form due to several issues: (1) limited novelty, (2) limited/unclear experimental validation, and (3) presentation issues.